# β-Aminobutyric Acid and Powdery Mildew Infection Enhanced the Activation of Defense-Related Genes and Salicylic Acid in Cucumber (*Cucumis sativus* L.)

**DOI:** 10.3390/genes14112087

**Published:** 2023-11-17

**Authors:** Ja-Yoon Kim, Hee-Wan Kang

**Affiliations:** 1Division of Horticultural Biotechnology, School of Biotechnology, Hankyong National University, Anseong 17579, Republic of Korea; xhfhcl@hknu.ac.kr; 2Institute of Genetic Engineering, Hankyong National University, Anseong 17579, Republic of Korea

**Keywords:** *Sphaerotheca fusca*, BABA, cucumber, defense genes

## Abstract

Powdery mildew disease, caused by *Sphaerotheca fusca*, is a major disease affecting cucumbers cultivated in greenhouses. This study was conducted to find defense genes induced by β-aminobutyric acid (BABA) and powdery mildew in cucumber. Disease severities of 25% and 5% were exhibited by the 2000 and 5000 mg/L BABA-treated cucumber, respectively. BABA did not affect the spore germination of the powdery mildew pathogen, showing that BABA is not an antifungal agent against the pathogen. In quantitative real-time PCR analysis, BABA-treated cucumber upregulated the transcriptional levels of the defense genes *CsPAL*, *CsPR3*, *CsPR1*, *CsLOX1*, *CsLOX23*, Cs *LecRK6.1*, *CsWRKY20*, and *Cupi4* in cucumber to maximum levels at 48 h, whereas *CsLecRK6.1* reached maximum expression after 24 h, and further, salicylic acid (SA) levels were significantly increased in BABA-treated cucumber plants. In addition, the cucumber infected with powdery mildew underwent a 1.6- to 47.3-fold enhancement in the defense genes *PAL*, *PR3*, *PR1*, *Lox1*, *Lox 23*, *LecRK6.1*, *WRKY20*, and *Cupi4* compared to heathy cucumber. These results suggest that the BABA-induced defense response is associated with SA signaling pathway-dependent systemic acquired resistance (SAR) in cucumber, which is involved in plant resistance mechanisms.

## 1. Introduction

Cucumber powdery mildew (CPM), caused by *S. fusca*, is a serious airborne disease affecting cucumbers. The disease is very difficult to manage because the conidia can attach to host plants, after which germination begins. One generation is completed in 5–6 days, and PM spreads very quickly among plants [1]. Particularly, the pathogen causes severe damage to domestically cultivated vegetables because the pathogen density increases during continuous cultivation in growing facilities. Although disease control has mainly relied on chemical fungicides, this approach has resulted in the emergence of drug-resistant pathogens. Alternatively, resistant plants can be developed as an effective, environmentally friendly, and economical method for controlling CPM.

Plants resist pathogenic bacteria and fungi to protect themselves from diseases. Induced resistance is a physiological “state of enhanced defensive capacity” elicited by specific environmental stimuli, enabling a plant to defend against subsequent biotic challenges [2]. Defense-related genes have been well-characterized in different plant hosts during infection with pathogenic and nonpathogenic fungi and are useful for evaluating the molecular mechanisms of disease resistance [3,4]. A previous study showed that five defense genes (*CsWRKY20*, *CsLecRK6.1*, *PR3*, *PR1-1a*, and *LOX1*) were expressed in cucumber leaves after inoculation with *Phytophthora melonis* [5]. Quantitative real-time polymerase chain reaction (qRT-PCR) showed that non-pathogenic *Fusarium oxysporium* CS-20 induced the expression of the defense genes *PR3*, *LOX1*, and *PAL1* in pre-inoculated cucumber roots, whereas the pathogenic fungus-mediated defense response was regulated by *PR1* and *PR3* [4]. The defense gene *LecRK-6.1* is also associated with resistance to PM in wheat, and the salicylic acid (SA) pathway contributes to enhanced resistance to virulent PM fungi [6].

The chemical elicitor BABA induces a systemic defense response against various plant pathogens [7]. BABA enhances resistance to *Botrytis cinerea* and *Pseudomonas syringae* in *Arabidopsis* and induces transcriptional expression of the *PR-1* gene, which is correlated with the SA-dependent systemic acquired resistance (SAR) system. Additionally, BABA treatment suppresses *Phytophthora* blight and induces defense gene expression in pepper [8,9]. The cucumber PM pathogen (*Sphaerotheca fuliginea*) and defense signal chemicals SA, abscisic acid, and methyl jasmonate were used to treat cucumber leaves, and it was revealed that 23 C-lipoxygenase-related genes (*CsLOXs*) in the cucumber genome were up- or downregulated in response to the signal chemicals [7]. SA and jasmonic acid (JA) transduce signals in the defense response to SAR and induce systemic resistance (ISR). The expression of defense genes in the SA and JA pathways is induced in BABA-treated plants [2,7,10]. It has been known the PR1 and PR3 genes encoding the pathogenicity-related protein (PR) are related to the SA signaling pathway [2,4]. In addition, ISR is usually triggered by nonpathogenic micro-organisms, and the related defense genes NPR1 and PAL, encoded by the PAL1, are activated by the JA/ethylene signaling pathway [4]. Ethylene and lipoxygenase (LOX), encoded by the LOX1 gene, are the first enzymes in the biosynthesis pathway of JA [4,7]. In addition, biochemical and enzymatic defense analyses suggested that BABA treatment significantly regulated catalase, guaiacol peroxidase, PAL activity, H_2_ O_2_, and lignin content [11]. Nevertheless, the expressional regulation of defense-related genes and the chemical signal response by PM pathogens and BABA still remain to be elucidated in cucumber plants.

This study was conducted to find the expression levels of defense-related genes in cucumber via BABA treatment and PM infection, and further, to elucidate whether BABA-treated cucumber plants produce chemical signals related to resistance responses to SAR or ISR.

## 2. Materials and Methods

### 2.1. Host Plant and Infection

The cucumber variety Baeckdadagi (Dongwon Nong San seed Co., Ltd., Seoul, Republic of Korea) was used as the host plant for PM. Seeds were sown in 48-pore trays filled with peat, and the seedlings were transplanted into plastic pots (12 cm) and grown for 4 weeks in a greenhouse. The CPM conidia were isolated from naturally infected cucumber leaves and suspended in distilled water containing 0.0003% Tween 20. A spore suspension of 4 × 10^6^ /mL was sprayed onto cucumber leaves and they were maintained in a greenhouse to monitor for disease occurrence. Distilled water (DW) was used as a control. The infection degree was evaluated based on a 0–5 scale, where 0 = no disease symptoms, 1 = less than 5% PM symptoms, 2 = 5.1–20% PM symptoms, 3 = 20.1–40% of plant disease, 4 = more than 40–60% diseased plant, and 5 = more than 61% PM symptoms. Disease severity was calculated using the method reported in [11] as follows: R = [Σ (a × b)/N × K] × 100%, where R is disease severity, a is the number of infected leaves rated, b is the numerical value of each grade, N is the total number of examined plants, and K is the highest degree of infection in the scale. Ten plants were used for each treatment, with three replicates per treatment.

### 2.2. Spore Germination

The fleshed conidia were isolated from cucumber PM lesions, spread onto 2% agar-coated glass slides, and covered with a cover glass. The agar blocks were incubated under fluorescent light for 48 h at 25 °C. The germination rate was measured for randomly selected conidia after two days.

### 2.3. qRT-PCR

An experiment was carried out to qualify the expression induction of defense genes on cucumber treated and infected with BABA and CPM. The primers targeting defense genes in cucumber used in qRT-PCR analysis are listed in Table 1. The cucumber seedlings grown in a greenhouse for four weeks were treated with 2000 and 5000 mg/L BABA via foliar spray. The leaves were collected at 0, 24, 48, and 72 h after the treatments. On the other hand, CPM-infected cucumber and heathy cucumber leaves were also used in the analysis. Total RNA was extracted from infected and BABA-treated cucumber leaves and control treatments using a Favorgen kit (Favorgen, Vienna, Austria) according to the manufacturer’s instructions. After treating the RNA samples with RNase-free DNase (Invitrogen Ambion DNase I (Waltham, MA, USA), 1 µg DNA-free RNA was used for first-strand cDNA synthesis with a HelixCript™ easy reverse transcriptase reagent (Nanohelix, Taejeon, Korea). The reaction mixture (20 μL) consisted of 10 μL of Dyne qPCR 2X PreMIX (DyneBio, Seongnam, Korea), 1000 ng/μL of cDNA, and 10 pmol of each primer. Activation was carried out, followed by 40 cycles of 95 °C for 15 s, 58 °C for 30 s, and 72 °C for 30 s. Relative gene expression was normalized to the level of mRNA using a LightCycler^®^ 96 System (Roche, Basel, Switzerland).

### 2.4. Extraction and Measurement of SA from Cucumber Leaves

The 2000 and 5000 mg/L BABA solutions were foliarly sprayed on the cucumber leaves. After 72 h, the leaves were collected and ground with liquid nitrogen. A total of 90 g of silicon dioxide and 30 mL of 90% methanol were added to 10 g of the ground leaves, and the supernatant was collected using a mortar bowl. The methanolic extracts were concentrated using a vacuum evaporator. The residues were resuspended in a mixture of 1mL of 5% trichloroacetic acid (Sigma-Aldrich, St. Louis, MO, USA) and 10 mL of 99.8% methanol (*w*/*v*). The volume of each extract was adjusted with distilled water to 50 mL and centrifuged at 8000× *g* (RCF) for 10 min. The supernatants of the extracts were used to measure the SA content using a liquid chromatograph–mass spectrometer (LCMS8050; Shimadzu, Kyoto, Japan) on a Kinetex C18 column (2.6 μm, 100 mm × 2.1 mm; Phenomenex, Torrance, CA, USA). The calibration curves were linear (R^2^ > 0.99) over a concentration range of 10–500 ng/mL with acceptable accuracy and precision.

### 2.5. Statistical Analysis

Data were analyzed using SIGMA PLOT 11.0 (http://www.sigmaplot.com, accessed on 12 November 2023). The significance of the data was evaluated using analysis of variance, followed by Tukey’s multiple range test (*p* < 0.05).

## 3. Results

### 3.1. BABA Inhibits Cucumber PM Disease

To investigate whether BABA protects against CPM, cucumber leaves were inoculated with PM spores. After 1 day, BABA was applied at concentrations of 2000 and 5000 mg/L and the disease severity was confirmed 11 days after inoculation. As shown in Figure 1A, 2000 and 5000 mg/L BABA-treated cucumber exhibited disease severity of 25% and 5%, respectively, whereas the water-treated cucumber leaves were severely infected, with a disease severity of 90%. In BABA-treated cucumbers, the PM was suppressed, and symptoms were largely absent, whereas in water-treated cucumber leaves, typical PM infection was observed on the surface of the cucumber leaves (Figure 1B). In addition, the inhibitory effect of BABA on PM spore germination was examined to determine whether BABA directly inhibits PM. PM spores treated with 2000 and 5000 mg/L BABA and water showed germination rates of 45%, 42%, and 40%, respectively (Figure 2A). This result suggests that BABA is not fungitoxic towards CPM spores and may induce disease resistance, as reported previously [2,10].

### 3.2. Defense Genes in Cucumber Were Upregulated by BABA and PM Infection

After treating cucumber plants with 2000 mg/L BABA, the expression levels of defense genes were quantified using qRT-PCR at 0, 24, 48, and 72 h. (Figure 3) The expression levels of defense genes in cucumber plants began increasing at 24 h after BABA treatment; except for the *LecRK6.1* gene, which showed the highest expression level at 24 h, the other defense genes evaluated showed the highest expression levels at 48 h, although these levels were decreased at 72 h. *PAL*, *PR3*, *PR1*, *Lox1*, *Lox 23*, *LecRK6.1*, *WRKY20*, and *Cupi4* were transcriptionally expressed at 2.7, 53.1, 25, 2.7, 2.8, 4.4, 0.6, and 21.6-fold higher levels compared to those in non-treated BABA cucumber plants at 48 h after BABA treatment. In addition, the relative expressions of defense genes were quantified in cucumbers infected with PM and treated with BABA. The transcriptional expression levels of defense genes in PM-infected and uninfected cucumbers were analyzed using qRT-PCR with a primer set targeting the cucumber disease resistance genes shown in Table 1. The transcriptional expression levels of the defense genes *PAL*, *PR3*, *PR1*, *Lox1*, *Lox23*, *LecRK6.1*, *WRKY20*, and *Cupi4* in infected cucumber were increased 3.4, 24.2, 7.8, 2.5, 3.9, 29.5, 1.6, and 47.3-fold compared to those in uninfected cucumber (Figure 4).

### 3.3. BABA Induces SA Accumulation in Cucumber

The defense genes PAL, PR1, and PR3 were upregulated at the transcriptional level. Defense genes are triggered through an SA-dependent signaling pathway, reflecting that BABA treatment induces SA accumulation in cucumbers [2,12]. To estimate whether BABA-treated cucumbers accumulated SA, the total SA content in BABA-treated cucumbers was determined using liquid chromatography–mass spectrometry. SA levels were increased 3.8–14-fold in 2000 and 5000 mg/L BABA-treated leaves compared with those in water-treated leaves (Figure 5). Another type of resistance, induced systemic resistance (ISR), is triggered by the JA/ET signaling pathway. It was confirmed that JA was accumulated in BABA-treated cucumbers; however, JA was not detected using liquid chromatography–mass spectrometry. Thus, this result suggests that BABA is associated with the SA signaling pathway-dependent SAR response in cucumber.

## 4. Discussion

*Sphaerotheca fuliginea*, the causal agent of PM disease, is an obligate parasite that cannot be cultured in artificial media and has a broad host range, particularly in cucurbit crops; the disease is problematic worldwide [13]. In the form of cleistothecia containing a few ascuses, *S. fuliginea* spends the winter on the remains of diseased plants and is the primary source of infection [13]. The pathogen spreads rapidly in cucumber cultivation facilities and is extremely difficult to control, leading to large amounts of damage. Although chemical drugs have been effectively used to control PM disease, their overuse can lead to the emergence of drug-resistant strains and environmental contamination problems. Therefore, PM disease-resistant plants should be developed to overcome these problems. Selecting useful genes identified through searches for disease resistance genes in cucumber plants can provide basic information for producing disease-resistant varieties. Elicitors and plant pathogens induce the expression of genes related to plant innate immunity and can be potentially used for profiling defense genes in plant genomes. [14].

In the present study, we assessed the expression patterns of defense genes in cucumber plants treated with BABA. BABA is a non-protein amino acid that rarely occurs in nature and participates in a wide range of activities as an inducer of resistance to a broad spectrum of plant pathogens such as viruses, bacteria, fungi, and nematodes [10]. Previously, different concentrations (0.5, 2, and 4 mM) of BABA were used to control spinach PM, and 4 mM BABA was the most effective against PM [11]. Additionally, 10–100 mM BABA was used to control lettuce downy mildew, with 10 mM BABA shown to be effective for controlling the disease [15]. We examined the protective effects of different BABA concentrations against PM. In this study, it was revealed that concentrations of more than 2000 mg/L BABA inhibited cucumber PM, indicating that a higher concentration of BABA than that reported previously is required to control cucumber PM.

The expression of the defense genes *PR3*, *PR1-1a*, *CsWRKY20*, *CsLecRK6.1*, and *Cupi4* was increased when cucumbers were artificially inoculated with *P. melonis*, which causes damping-off disease [2]. In addition, *Fusarium oxysporum*-infected cucumber shows strongly enhanced expression of *PR3*, *LOX1*, and *NPR1* [4], and it was recently reported that phenolic compounds and flavonoid-related metabolites accumulate in the cucumber PM-resistant cultivar BK following infection with *S. fuliginea* [16]. However, the expression patterns of different defense genes in cucumbers caused by PM infection have not been defined. In this study, the expression levels of the defense genes *PAL*, *PR3*, *PR1*, *Lox1*, *Lox 23*, *LecRK6.1*, *WRKY20*, and *Cupi4* were significantly increased 1.6–47.3-fold under biotic stress due to PM infection. This is the first study to provide information on the transcriptional expression levels of different defense genes in both BABA-treated and PM-infected cucumber plants. The defense genes induced in cucumber plants showed expression values similar to those previously reported for other plant pathogens. BABA did not directly affect the germination of PM; therefore, it did not exert direct antifungal action against the PM pathogen, suggesting that BABA participates in defense mechanisms by inducing disease resistance in plants.

The receptors in the plant cell membrane recognize pathogen-associated molecular patterns (PAMP) and activate the plant immune system, including the induction of disease resistance-related gene expression [2]. In this study, the upregulated expression of various defense genes in PM-infected cucumbers may be attributed to the triggering of disease resistance mechanisms related to the innate immune system in the cucumber genome. *PR1*, *PR3*, and lipoxygenases (*LOX*) are strongly induced when plants respond to pathogen infection [2,17]. Pathogenesis-related proteins (PRs) are among the most commonly induced proteins in plant defense mechanisms and play important roles in plant immunity [16]. *PR1* strongly inhibits *Phytophthora infestans* in potatoes [18], and *PR1* expression in plant cells is a useful molecular marker for the SA-dependent SAR signaling pathway and may be related to its putative direct antimicrobial action [19]. Furthermore, *PR3* (endochitinase), which hydrolyzes chitin components via biotic and abiotic stress, is often used as a marker for SA-dependent SAR signaling [2,8,20]. Chitinase degrades the fungal cell wall and degradable products, chitin oligomers, which can serve as elicitors of plant disease resistance [20]. *Fusarium* wilt pathogen (*Fusarium oysporium* f.sp. *cucumerium*)-responsive chitinase genes were profiled using comparative transcriptome analysis, revealing six genes that were significantly upregulated after pathogen infection. Additionally, *LOX* pathways are crucial for lipid peroxidation during defense responses to infection and inhibit pathogen growth [21]. *LOX* genes are upregulated in cucumber leaves following infection with the PM pathogens *S. fuliginea*, SA, JA, and ABA [10]. *PAL* generates cinnamic acid by catalyzing the non-oxidative deamination of phenylalanine to *trans*-cinnamate. Given that phenylpropanoids derived from cinnamic acid serve as precursors for a range of phenolic compounds, they play a vital role in the biosynthesis of SA, which is an essential signal in the SAR response of plants [22].

In this study, *LecRK6.1* showed the highest expression level among the defense genes in PM-infected and BABA-treated cucumber leaves. Interestingly, the highest expression level of *LecRK6.1* was observed at 24 h after BABA treatment rather than at 48 h after BABA treatment. The *LecRK* family of genes plays an important role under biotic and abiotic stresses in plants by inducing plant innate immunity [1,23]. In cucumbers, *CsLecRK6.1* is induced by *P. melonis* and *Phytophthora capsici* in resistant cultivars [2,23]. In cucumber plants, the expression of *CsLecRK6.1* was increased by *P. melonis* and *P. capsici*. *LecRK-V* confers broad resistance to wheat PM through an SA pathway-dependent disease resistance mechanism [6]. Resistance is activated in plant cells through infection with plant pathogens, which is determined by an intricate network of signaling pathways involving innate immunity and resistance machinery [2]. *WRKY* is a plant-specific transcription factor with an important role in plant defense. *WRKY* enhances resistance to plant pathogens by increasing the expression levels of PR genes and inducing the accumulation of phytoalexins [23,24]. In addition, *WRKY* transcription factors positively regulate plant immunity to pathogens associated with H_2_O_2_ production, and a hypersensitive response mimics cell death and the activation of phytohormone-mediated signaling pathways [25]. However, we found that the transcriptional expression of *WRKY20* was 1.6 times higher in PM cucumber leaves, whereas its expression was downregulated in BABA-treated cucumber leaves. These results suggest that cucumber *WRKY20* gene expression was not sufficiently induced by BABA or PM infection. *Cupi4*, which is homologous to a pathogen-inducible protein, has been isolated from cucumbers infected with bacterial pathogens and characterized as a chemical inducer of SAR, such as SA, and pathogenic bacteria, viruses, and fungal treatment in cucumbers led to upregulation of the Cupi4 transcript [26]. It was predicted that the gene product was associated with SAR. In this study, the expression of the Cupi4 transcript was increased 21.6–47.3-fold at 48 h in BABA-treated and PM-infected cucumbers, respectively. PAL, PR3, PR1, Lox1, Lox 23, LecRK6.1, WRKY20, and Cupi4 are associated with SA signaling pathway-dependent SAR. They showed relatively high expression levels in BABA-treated cucumber plants. Accordingly, we examined whether SA accumulation occurred in BABA-treated cucumbers. As expected, 14.1 ng/mL and 3.84 ng/mL SA were detected in the 5000 and 2000 mg/L BABA-treated cucumbers, whereas 0.92 µg/mL SA was detected in untreated cucumbers. This result indicates that BABA treatment led to the accumulation of SA, suggesting that SA is involved in the SAR mechanism. These results show that BABA may be used as an inducer of SAR against cucumber PM, and that BABA and PM infection-induced defense genes can be used as target genes for developing PM-resistant plants. Although some studies have been conducted to induce the expression of resistance genes in cucumbers following infection with various plant pathogens, few studies have focused on the expression of resistance genes targeting cucumber PM. We focused on the expression of defense genes induced by BABA treatment and cucumber PM, targeting previously reported genes in plants, including cucumber. In future, it is expected that BABA- and PM-induced additional defense genes will be able to be identified in cucumber plants at the genome level through RNA-seq technology.

## 5. Conclusions

We found that the transcriptional expression of defense genes in cucumbers was induced by PM infection and BABA treatment. BABA had a protective effect of more than 80% against PM disease and did not affect PM spore germination, indicating that it did not directly inhibit pathogens. In BABA-treated and PM-infected cucumbers, the transcripts of the defense genes *PAL*, *PR3*, *PR1*, *LOX1*, *LOX23*, *LecRK6.1*, *WRKY20*, and *Cupi4* were significantly increased, demonstrating their association with SA-dependent SAR resistance. This study provides a foundation for functional studies of cucumber plant defense responses and signaling pathways against PM.

## Figures and Tables

**Figure 1 genes-14-02087-f001:**
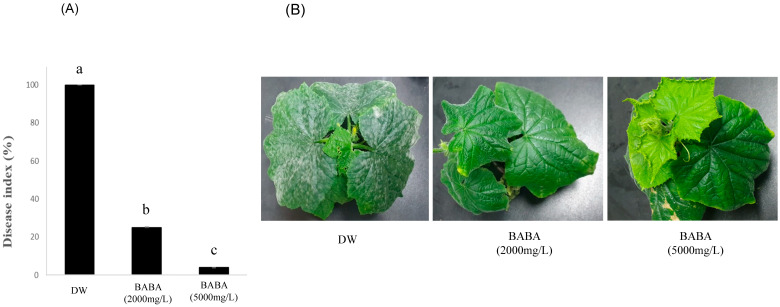
Protective effect of β-aminobutyric acid (BABA) against cucumber powdery mildew. (**A**) Powdery mildew disease index and (**B**) disease index observed at 14 days after inoculation. The value represents the mean disease index ± standard deviation. The experiment was repeated three times. DW, distilled water. Different letters indicate significant differences between treatments (*p* < 0.05 according to Duncan’s multiple range test).

**Figure 2 genes-14-02087-f002:**
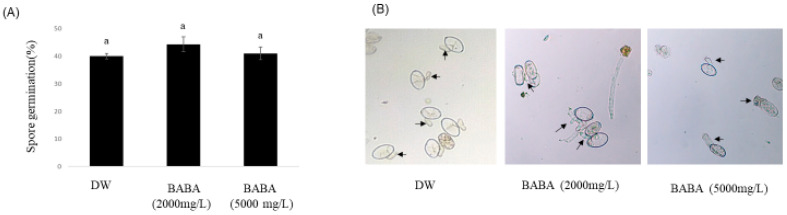
Inhibition rate of spore germination on different concentrations of β-aminobutyric acid against cucumber powdery mildew. (**A**) Spore germination rate following treatment with different BABA concentrations and (**B**) conidial germination of powdery mildew pathogen derived from cucumber. Letter “a” indicates significant differences between treatments (*p* < 0.05 according to Duncan’s multiple range test). The arrowheads indicate germ tubes formed from the spores.

**Figure 3 genes-14-02087-f003:**
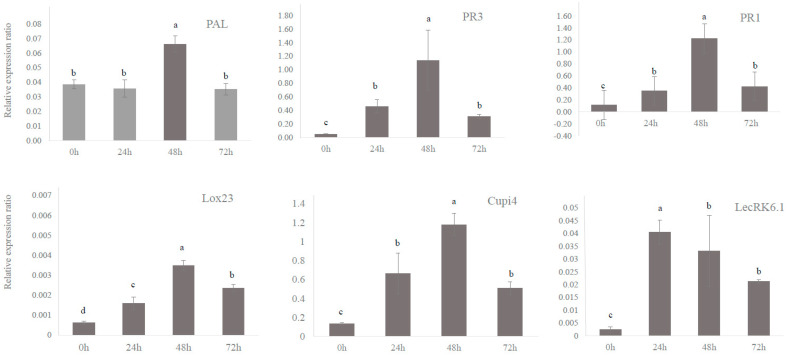
Expression levels of defense genes in cucumber at different hours after treatment with BABA (2000 mg/L). BABA (2000 mg/L) was administered to cucumber leaves with different durations (0, 24, 48, and 72 h). Total RNA was extracted from each treated cucumber leaf sample and subjected to qRT-PCR using primers targeting the defense genes shown in Table 1. Each gene expression was normalized to the reference gene, *CsActin*. The expression value is the average of three replications, and the bar indicates the standard deviation. Different letters indicate significant differences between treatments (*p* < 0.05 according to Duncan’s multiple range tests).

**Figure 4 genes-14-02087-f004:**
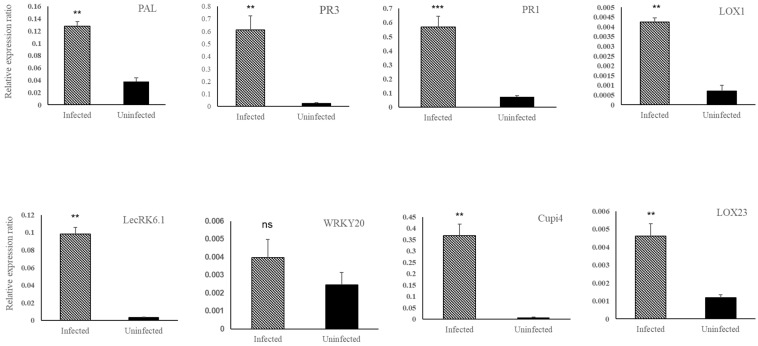
Quantitative real-time PCR (qRT-PCR) analysis of defense genes in powdery mildew-infected and uninfected cucumber leaves. The expression of each gene was normalized to that of the reference gene, *Actin*. The expression value is shown as the average of three replicates, and the bar indicates the standard deviation. **, *** and ns indicate significant differences and no significance between treatments (*p* < 0.05 according to Duncan’s multiple range test).

**Figure 5 genes-14-02087-f005:**
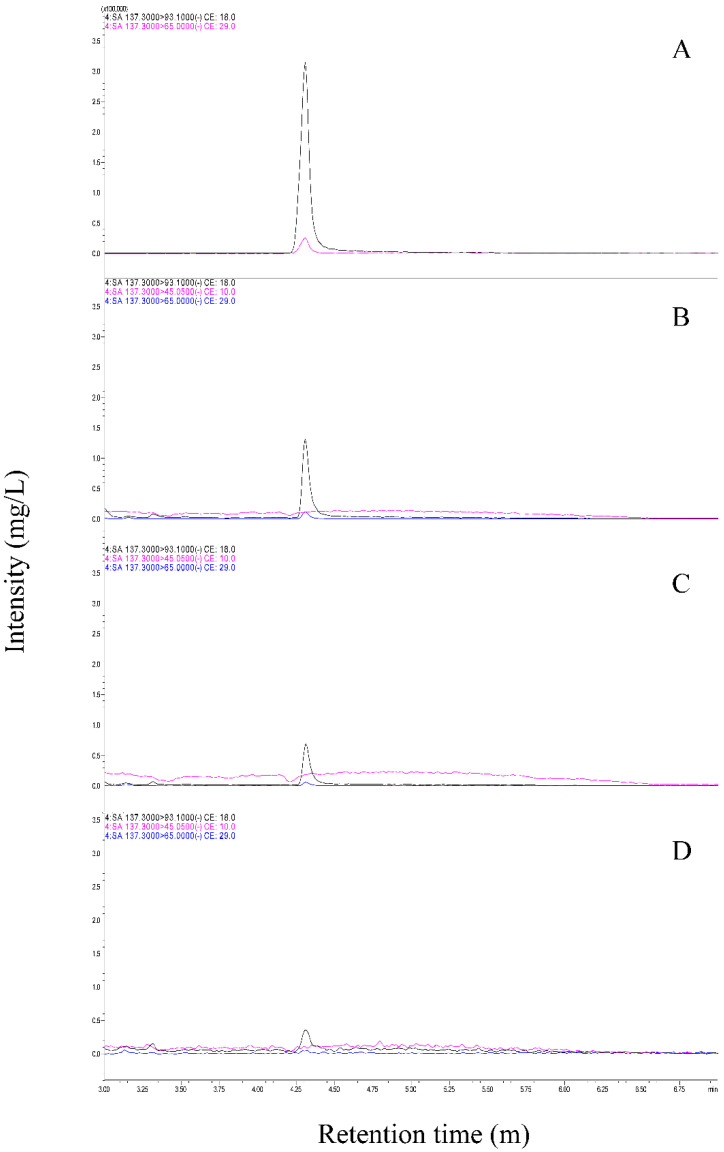
Accumulation of salicylic acid (SA) in BABA-treated cucumber. (**A**) SA standard. Cucumber plants were treated with BABA 5000 mg/L (**B**), 2000 mg/L (**C**) and distilled water (**D**). SA 0.92 mg/L, 3.84 mg/L and 14.84 mg/L were detected in BABA treated B, C and D samples.

**Table 1 genes-14-02087-t001:** Primer sequences targeting defense genes in cucumber.

Genes	Primer Sequence
*CsActin*	F: 5′-TCG TGC TGG ATT CTG GTG-3′
	R: 5′-GGC AGT GGT GGT GAA CAT-3′
*CsPAL*	F: 5′-AAA CAC GTC GGA TAA ATA TGG CTT-3′
	R: 5′-CAT CCA TTC AGG CGT TCC AG-3′
*CsPR3*	F: 5′-CAC TGC AAC CCT GAC AAC AAC G-3′
	R:5′-AAG TGG CCT GGA ATC CGA CTG-3′
*CsPR1*	F: 5′-CTC AAG ACT TCG TCG GTG TCC A-3′
	R: CGC CAG AGT TCA CTA GCC TAC
*CsLOX1*	F: 5′-TCT TTG CTT CAG GGT ATC AC-3′
	R: 5′-GCA AAT TCT TCA TCA CTA CTC C-3′
*LOX23*	F: 5′-TGC CTC CAA CAC CTT CTT CAA-3′
	R: 5′-CTT CCA TAT CAA ATC GCC ACA-3′
*CsLecRK6.1*	F: 5′-CGA CCA CAA CGA AAT GTC ACA C-3′
	R: 5′-TTT CTT CCA CAC GCC ACT TCC-3′
*CsWRKY20*	F: 5′-GAA ATA ACG TAC AGA GGG AAG C-3′
	R: 5′-CAG GTG CTG TTT GTT GGT TAT G-3′
*Cupi4*	F: 5′-TCA CTG TGG TGT GTG CTC TC-3′
	R:—ACT CAA GCC ATT GCC TTC CA-3′

## Data Availability

Data are contained within the article.

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
