# Peer review of "β-Aminobutyric Acid and Powdery Mildew Infection Enhanced the Activation of Defense-Related Genes and Salicylic Acid in Cucumber (Cucumis sativus L.)"

_genes, 2023, doi:10.3390/genes14112087_

Round 1
Reviewer 1 Report
Comments and Suggestions for Authors
The authors assessed the expression patterns of defense genes in cucumber plants treated with 2000 and 5000 mg/L BABA. They found that BABA did not effect spore germination of the powdery mildew pathogen, and BABA treated cucumber upregulated transcrip-16 tional level of the defense genes CsPAL, CsPR3,CsPR1, CsLOX1, CsLOX23, Cs LecRK6.1, CsWRKY20, 17 and Cupi4 in cucumber to maximum levels at 48 h, whereas CsLecRK6.1 reached maximum expression after 24 h and futher salicylic acid (SA) levels were significantly increased in BABA-treated 19 cucumber plants. The work is comparatively superficial and the content is currently not enough for publication in genes. And also I feel it is not fit for the "Plant Genecics and Genomics" section.
1. Line 142, when the BABA was applied to cucumber leaves, how long after the PM infection.
2. I suggest the authors to perform RNA-seq to dig more genes and pathways involved in BABA-mediated PM-resistance.
3. In Fig 1b, why there is no photo of BABA (5000mg/L)?
4. Fig 2b showed the mildew pathogen dericed from cucumber with DW or BABA treatment?
5. Fig 5 is too vague to read.
Author Response
Thank you for your comments and suggestions on our paper. The authors will sincerely respond to your comments and make corrections.
- Line 142, when the BABA was applied to cucumber leaves, how long after the PM infection.
Corrected: the following sentence was inserted in line 142; the disease severity was confirmed 11 days after inoculation
- I suggest the authors to perform RNA-seq to dig more genes and pathways involved in BABA-mediated PM-resistance.
Corrected: We described in lines 324-325 for your comment.
- In Fig 1b, why there is no photo of BABA (5000mg/L)?
Corrected
: the related photos were added in Fig. 1
- Fig 2b showed the mildew pathogen dericed from cucumber with DW or BABA treatment?
Corrected : The related photos were added in Fig.2
- Fig 5 is too vague to read.
Corrected : you can find the revised Fig. 5.
Reviewer 2 Report
Comments and Suggestions for Authors
Article
DL-3-Aminobutyric Acid and Powdery Mildew Infection Enhanced the Activation of Defense Related Genes and Salicylic Acid in Cucumber (Cucumis sativus L.)
A brief summary
The topic is extremely important, interesting and in keeping with current trends of ‘green’ horticulture production techniques. The research is properly designed, the results interesting. Unfortunately, the paper contains a number of inaccuracies, mistakes, inconsistencies and typos.
Broad comments
1. If abbreviations have been used, they should be used consistently, in the text and in the diagrams (for example, in the description of Fig. 2.). In order not to explain them every now and then (line 65 vs line 70, the description of Figure 3.), a summary of the abbreviations used can be included. In one place the tested compound (BABA) is described as "DL-3-Aminobutyric Acid" (Title, line 51, Figure 1 and 5), in another "DL-β-aminobutyric acid" (Figures 2 and 3).
2. Units should be standardised. Why are doses given as mg/L (e.g. Figure 2) in one place and as ppm (e.g. 167) in another?
3. It seems that scientific papers should be written in the passive voice. Here, the authors are probably very keen to highlight their achievements.
4. Typos need to be corrected, spaces inserted or removed, etc.
Specific comments
The keywords are not well chosen. Instead of "cucumber"; "powdery mildew"; "infection"; "DL-3-amino-butyric acid" (they are already in the title) I suggest "Sphaerotheca fusca"; "BABA", "spore germination" and "plant defence response".
Line 41. The phrase "non-pathogenic pathogens" is quite unfortunate.
Line 128. In what units is the centrifuge speed given?
Line 130. C18 not c18
Figure 1 and 2. No results of statistical analysis and no description of the statistical test.
Figure 3. It seems that the correctness of the homogenic groups shown should be checked. For example, in chart PR1 there is a 'bc' group and no 'c' group. The homogenous groups in the LecRK6.1 chart also look suspicious.
Figure 4. It would be more appropriate to mark significant differences with an asterisk rather than with homogeneous groups if only two levels of a factor are being studied.
Figure 5 is missing. The chromatograms in its place are also interesting and worth showing. They are probably wrongly described, however, because the bands (peaks) at the same retention time have different names. Were there no other peaks? The drawings are low resolution and poorly readable.
Lines 229-232. This sentence is a repetition of the sentence in lines 37-40. Interestingly, once the authors referred to source [2] and then to [15]. Letter S in is missing in the source [15] title.
Author Response
Broad comments
- If abbreviations have been used, they should be used consistently, in the text and in the diagrams (for example, in the description of Fig. 2.). In order not to explain them every now and then (line 65 vs line 70, the description of Figure 3.), a summary of the abbreviations used can be included. In one place the tested compound (BABA) is described as "DL-3-Aminobutyric Acid" (Title, line 51, Figure 1 and 5), in another "DL-β-aminobutyric acid" (Figures 2 and 3).
Corrected: DL β-aminobutyric acid was applied in this paper.
- Units should be standardised. Why are doses given as mg/L (e.g. Figure 2) in one place and as ppm (e.g. 167) in another?
Corrected: ppm mg/L
- It seems that scientific papers should be written in the passive voice. Here, the authors are probably very keen to highlight their achievements.
Corrected
- Typos need to be corrected, spaces inserted or removed, etc.
Corrected
Specific comments
The keywords are not well chosen. Instead of "cucumber"; "powdery mildew"; "infection"; "DL-3-amino-butyric acid" (they are already in the title) I suggest "Sphaerotheca fusca"; "BABA", "spore germination" and "plant defence response".
Corrected: "Sphaerotheca fusca"; "BABA", cucumber, "defense genes",
Line 41. The phrase "non-pathogenic pathogens" is quite unfortunate.
Corrected: nonpathogenic fungi
Line 128. In what units is the centrifuge speed given?
Corrected: at 8,000 x g
Line 130. C18 not c18
Corrected: C18
Figure 1 and 2. No results of statistical analysis and no description of the statistical test.
Corrected
: Different letters indicate significant differences between treatments (P < 0.05 according to Duncan’s multiple test).[ìœ 1]
Figure 3. It seems that the correctness of the homogenic groups shown should be checked. For example, in chart PR1 there is a 'bc' group and no 'c' group. The homogenous groups in the LecRK6.1 chart also look suspicious.
Corrected
Figure 4. It would be more appropriate to mark significant differences with an asterisk rather than with homogeneous groups if only two levels of a factor are being studied.
Corrected
Figure 5 is missing. The chromatograms in its place are also interesting and worth showing. They are probably wrongly described, however, because the bands (peaks) at the same retention time have different names. Were there no other peaks? The drawings are low resolution and poorly readable.
Corrected: the Fig.5 was revised with new photo of high resolution.
Lines 229-232. This sentence is a repetition of the sentence in lines 37-40. Interestingly, once the authors referred to source [2] and then to [15]. Letter S in is missing in the source [15] title.
Corrected: Revised sentence (Elicitors and plant pathogens induce the expression of genes related to plant innate immunity and can be potentially used for profiling defense genes in plant genomes) was in 232-233 lines.
Round 2
Reviewer 1 Report
Comments and Suggestions for Authors
The authors have solved all my questions.
However, the language should be improved. There are some errors in the revised manuscript.
In figure 1 and 2: There is a space between number and unit, e.g 2000 mg/L, not 2000mg/L
Line 199: analyses, not analysis
Line 207: induces, not iduces
Author Response
Dear Reviewer 1
Thank you for your comments. We have revised accordingly.
Best Regards
Reviewer 2 Report
Comments and Suggestions for Authors
The changes and amendments made have significantly improved the quality of the work. However, Figure 5 remains to be discussed. The changes made by the authors have resulted in much better readability of the chromatograms. The title of the figure refers to the accumulation of salicylic acid in the plant material studied. The chromatograms show the decreasing band corresponding to the acid and the amount of this compound. How much acid was in the standard? Why does the description mention the standard deviation and the description of the results of the statistical test and these data are missing from the chromatograms? Perhaps these data should be presented in a separate table or graph?
After clarifying this point, the article, seems suitable for publication.
Author Response
Dear Reviewer 2
Thank you for your kind comments. I will respond to the comments you presented.
1. How much acid was in the standard.
- The concentration of SA standard is 0.1 mg/L and it was referred in the content of Fig. 5.
2. Why does the description mention the standard deviation and the description of the results of the statistical test and these data are missing from the chromatograms? Perhaps these data should be presented in a separate table or graph?
- The data with the standard deviation and the statistical test was presented in Fig. 5A as a graph. Additionally, you can see the corrected SA chromatogram in Fig. 5B . The graph and SA chromatogram were attached in file.